# VER🗣: LEARNING NATURAL LANGUAGE REPRESENTATIONS FOR VERBALIZING ENTITIES AND RELATIONS

## ABSTRACT

Entities and relationships between entities are vital in the real world. Essentially, we understand the world by understanding entities and relations. For instance, to understand a field, e.g., *computer science*, we need to understand the relevant concepts, e.g., *machine learning*, and the relationships between concepts, e.g., *machine learning* and *artificial intelligence*. To understand a person, we should first know who he/she is and how he/she is related to others. To understand entities and relations, humans may refer to natural language descriptions. For instance, when learning a new scientific term, people usually start by reading its definition in dictionaries or encyclopedias. To know the relationship between two entities, humans tend to create a sentence to connect them. In this paper, we propose **VER🗣**: A Unified Model for **V**erbalizing **E**ntities and **R**elations. Specifically, we attempt to build a system that takes any entity or entity set as input and generates a sentence to represent entities and relations, named "*natural language representation*". Extensive experiments demonstrate that our model can generate high-quality sentences describing entities and entity relationships and facilitate various tasks on entities and relations, including definition modeling, relation modeling, and generative commonsense reasoning.[1]

## 1 INTRODUCTION

*What is X? What is the relationship between X and Y?* We come up with these questions almost every day. When we come across a new term, e.g., *twin prime*, we usually refer to its definition to understand it, i.e., "*A **twin prime** is a prime number that is either 2 less or 2 more than another prime number*". To express the understanding about relationship between entities (e.g., *carbon dioxide* and *water*), we create a sentence to represent their relationship: "***Carbon dioxide** is soluble in **water***". Basically, we understand entities and relations by "*verbalizing*" them. Verbalizing entities and relations also tests our knowledge about entities and relations. Literally, by verbalizing entities and relations, we understand the world.

Similarly, *do machines have the ability to verbalize entities and relations? Can machines learn about entities and relations from verbalizing them?* The answer is "*Yes*". Recent studies show that by giving the surface name of an entity (and its context), models (after training) can generate coherent sentences to represent it, i.e., **definition modeling** (Noraset et al., 2017; Gadetsky et al., 2018; Bevilacqua et al., 2020; August et al., 2022; Huang et al., 2021b; Gardner et al., 2022), and by giving the surface names of a pair of entities, machines can generate coherent sentences describing their relationships, i.e., **(open) relation modeling** (Huang et al., 2022a;b). However, verbalizing entities requires understanding relationships between entities, and verbalizing entity relationships requires understanding entities themselves, while existing works deal with entity and relation verbalization separately, ignoring the connections between them.

Besides, recent works (Devlin et al., 2019; Lewis et al., 2020; Radford et al., 2019; Brown et al., 2020) have shown that large language models pre-trained with self-supervised objectives can equip

---

[1] We release the VER-base model on this anonymous repo `https://osf.io/7csnf/?view_only= 91ec67e05bd44f998d71e63d9cdd25a4`. VER-large and the pre-training data will be released as open-source after the review process (since the anonymous repo has a space limitation).

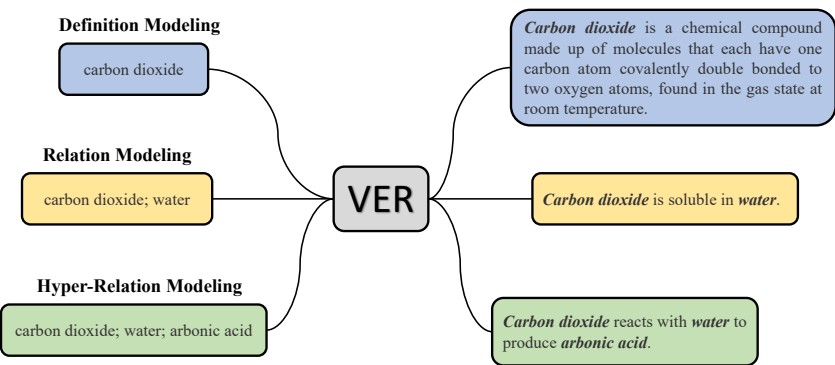

Figure 1: A diagram of VER. We feed the model with entity(s) and train it to reconstruct sentences containing all the entities. This allows us to use a single model to better "*verbalize*" entities and complex entity relationships.

the model with a significant amount of knowledge (Petroni et al., 2019; Roberts et al., 2020) and achieve substantial gains after fine-tuning on a specific task. Can we continually pre-train the models with pre-training objectives on entities and relations to enhance their ability on verbalizing entities and relations? In this way, the model can be easier and better adapted to specific tasks on entities and relations and even be used without additional training.

Therefore, we aim to solve entity and relation verbalization in a unified form and pre-train a model for entity and relation understanding. Essentially, definition modeling and relation modeling can be unified as an "entity(s) → sentence" task, i.e., given a set of entities, generating a sentence describing the entities and their relationships. When the size of the set is 1, it is equivalent to definition modeling, and when the size of the set is 2, it is equivalent to relation modeling. By defining the task in this form, we can even model more complex relationships among entities since entity relationships can go beyond pairwise (Bretto, 2013), named **hyper-relation modeling**, e.g., {*carbon dioxide*, *water*, *arbonic acid*} → "***Carbon dioxide*** *reacts with* ***water*** *to produce* ***arbonic acid***". Based on this, we propose **VER**: A Unified Model for **V**erbalizing **E**ntities and **R**elations (Figure 1). Specifically, we pre-train models by forming a self-supervised text reconstruction task: given an entity or a set of entities, reconstruct the original sentences (e.g., a definition or a relation description) containing them in the training corpus. In this way, the models acquire knowledge about entities and relations and learn to connect entities to a meaningful coherent sentence.

From the perspective of representation learning, we propose to learn the "*natural language representations*" of entities and relations. Compared to latent vector representations, natural language representations are more *interpretable* since humans can understand the representations by reading the texts, while hidden representations are difficult to interpret. Compared to structural representations with pre-specified rules, e.g., a sub-knowledge graph, natural language representations are more *open* owing to the flexibility of free texts.

Experiments on six datasets demonstrate the superiority of our model in verbalizing entities and relations. Especially in low-resource settings, our model can achieve significantly better results than BART (Lewis et al., 2020) on definition modeling, relation modeling, and hyper-relation modeling (generative commonsense reasoning (Lin et al., 2020)). In addition, the performance of VER without additional training is impressive, making itself a potential knowledge source of entities and relations, which may benefit tasks on entities and relations such as entity typing Ren et al. (2016), relation extraction (Bach & Badaskar, 2007), and knowledge graph completion (Lin et al., 2015).

The main contributions of our work are summarized as follows:

- We connect definition modeling, relation modeling, and hyper-relation modeling in a unified form;
- We pre-train VER on a large training data by forming the "entity(s) → sentence" reconstruction task, which makes VER a useful tool for learning natural language representations of entities and relations;
- Extensive experiments demonstrate our model can achieve better results in verbalizing entities and relations, especially in low-resource settings.

## 2  BACKGROUND AND FORMULATIONS

**Definition Modeling**. Definition modeling aims to generate definitions of entities, which can be formulated as a conditioned sequence generation task. For instance, given *twin prime*, the expected output is the definition of *twin prime*: "A *twin prime* is a prime number that is either 2 less or 2 more than another prime number". We follow the standard sequence-to-sequence formulation in (Noraset et al., 2017; Huang et al., 2021b): given entity $x$, the probability of the generated definition $s = [s_1, \ldots, s_m]$ is computed auto-regressively:

$$P(s|x) = \prod_{i=1}^{m} P(s_i|s_0, s_1, \ldots, s_{i-1}, x), \tag{1}$$

where $m$ is the length of $s$, $s_i$ is the $i$th token of $s$, and $s_0$ is a special start token.

**(Open) Relation Modeling**. Relation modeling attempts to generate coherent and meaningful sentences describing relationships between entities, where types of relations are not pre-specified, i.e., in an "*open*" setting (Huang et al., 2022a). For example, for *carbon dioxide* and *water*, their relationship can be described as "*Carbon dioxide* is soluble in *water*." For *machine learning* and *algorithm*, the expected output could be "*Machine learning* explores the study and construction of *algorithms* that can learn from and make predictions on data." Formally, given entity pair $(x, y)$, the probability of the generated relation description $s = [s_1, \ldots, s_m]$ is calculated as:

$$P(s|x, y) = \prod_{i=1}^{m} P(s_i|s_0, s_1, \ldots, s_{i-1}, x, y). \tag{2}$$

**Hyper-Relation Modeling (Unified Form)**. Previous works mainly focus on verbalizing single entities or entity pairs. However, in the real world, relationships between entities can be more complex – beyond pairwise, named "hyper" relationships (Bretto, 2013; Tu et al., 2018; Huang et al., 2019). For example, "*carbon dioxide* reacts with *water* to produce *carbonic acid*". Here, there are tuplewise relationships among *carbon dioxide*, *water*, and *carbonic acid*. Verbalization of hyper relationships was initially investigated in (Lin et al., 2020) but was limited to commonsense concepts, and their outputs are simple short sentences describing everyday scenarios containing the given concepts. We attempt to model and verbalize more general complex "hyper" relationships among entities and find a unified framework to combine single entities (1 entity), pairwise relationships (2 entities), and "hyper" relationships ($\geq 3$ entities). Combining with definition modeling and relation modeling, we adopt the following unified form:

$$P(s|\mathcal{E}) = \prod_{i=1}^{m} P(s_i|s_0, s_1, \ldots, s_{i-1}, \mathcal{E}), \tag{3}$$

where $\mathcal{E}$ is the entity set and $|\mathcal{E}| \geq 1$.

**Natural Language Representation**. Natural language representation aims to represent entities and relations with text descriptions. Natural language representation is *interpretable* since texts are readable. Natural language representation is *open* since most things can be described in natural language. Based on the unified form above, we define natural language representation of entities and relations as $r(\mathcal{E}) = s$. Here we should notice that the surface name of entities does not provide much semantic knowledge; therefore, producing natural language representations requires the model to generate coherent sentences describing entities and relations based on knowledge stored in the parameters of the model the reasoning ability of the model.

## 3  VER🔊: VERBALIZING ENTITIES AND RELATIONS

To verbalize an entity, we are likely to connect it to other entities, which requires knowledge about entity relationships. To understand entity relationships, we need to know about entities first. Based on this, we attempt to verbalize entities and relations in a unified form and propose **VER🔊**: A Unified Model for **V**erbalizing **E**ntities and **R**elations. We first create a large dataset with the formulation in Eq. (3), and pre-train a model on this dataset, which equips the model with a significant amount of knowledge about entities and relations and enables the model to generate coherent

and meaningful sentences connecting the entities. The model can be further fine-tuned on specific datasets, e.g., definition modeling, relation modeling, and generative commonsense reasoning, to achieve better performance on specific tasks.

## 3.1 DATA

We prepare the pre-training data with Wikipedia. Wikipedia is a large encyclopedia containing a huge number of entities. Wikipedia is well maintained and the content is generally of high quality. We extract entity sets and sentences from Wikipedia. Specifically, we use the 2022-08-01 dump[2] of English Wikipedia. For each page, we extract the plain text by WikiExtractor[3]. And we use the neuralcoref (Clark & Manning, 2016) coreference resolution tool in `spaCy`[4] to preprocess the documents. Since we would like the model to capture the main characteristics of entities and relations, we take the first 5 sentences from each page (those sentences are usually definitional sentences or sentences expressing entity relationships). To identify entities, we utilize the hyperlinks in each Wikipedia page to build a local mention-entity mapping. And then, we process each sentence and extract the corresponding entity set based on the mention-entity mapping. In this way, we build mapping $\mathcal{E} \rightarrow s$, e.g., "{*Data mining, data sets, machine learning, statistics, database systems*} $\rightarrow$ *Data mining is the process of extracting and discovering patterns in large data sets involving methods at the intersection of machine learning, statistics, and database systems.*" Since for a single entity, we prefer the model to generate a definition-like sentence rather than a random sentence including it, we collect the first sentence on each page and collect the input-output pair as "{*[page title]*} $\rightarrow$ *1st sentence*", e.g., "{*deep learning*} $\rightarrow$ *Deep learning is part of a broader family of machine learning methods based on artificial neural networks with representation learning.*" We filter out input-output pairs where $|\mathcal{E}| = 1$ and $s \neq$ *1st sentence*. We keep out pages appearing in the validation and test sets of (Huang et al., 2021b; 2022a) and filter out entity sets appearing in the datasets of (Huang et al., 2021b; August et al., 2022; Huang et al., 2022a; August et al., 2022; Lin et al., 2020). The number of training examples with different sizes of entity sets is shown in Figure 2.

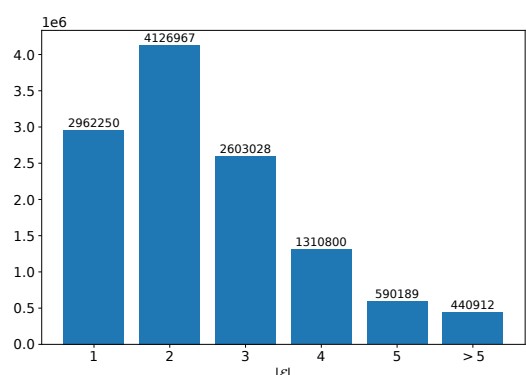

Figure 2: Statistics of the pre-training data.

## 3.2 MODEL

At a high level, we pre-train a model by training it to reconstruct target sentences conditioned on the entity set with the pre-training data. Specifically, we continually pre-train BART (Lewis et al., 2020) on the data constructed in Section 3.1. BART adopts a transformer-based encoder-decoder architecture with input text fed to the encoder and output text produced by the decoder. For our continual pre-training, we encode entity set $\mathcal{E} = \{e_1, e_2, \ldots, e_{|\mathcal{E}|}\}$ to sequence "$e_1$; $e_2$; $\ldots$; $e_{|\mathcal{E}|}$", e.g., {*carbon dioxide, water, carbonic acid*} to "`carbon dioxide; water; carbonic acid`". Here we keep the order of entities as the order they appear in the sentence. We choose this design because different orders may correspond to different natural language representations (e.g., the representations are different when an entity is used as subject vs. object). We would like to mention that although we keep the order here, the model can deal with inputs with random entity orders after fine-tuning (e.g., CommonGen (Lin et al., 2020) as shown in Section 4.4). We train two versions of the model: *VER-base* with 6 layers in the encoder and decoder, and *VER-large* with 12 layers in each, corresponding to BART-based and BART-large respectively.

---

[2]`https://dumps.wikimedia.org/enwiki/20220801/`
[3]`https://github.com/attardi/wikiextractor`
[4]`https://spacy.io/`

### 3.3 TRAINING PROCESS

We pre-train VER-large and VER-base with the `fairseq` library[5]. We use Adam with $\beta_1 = 0.9$, $\beta_2 = 0.999$, and $\epsilon = 10^{-8}$, and set the clip threshold of gradients as $0.1$. Both models use weight decay of $0.001$ and dropout of $0.1$. We set the learning rate as $5 \times 10^{-5}$ and use batch size of $1,024$ tokens, updating every $16$ iterations. We set the number of warmup steps as $1,000$. We use a small validation set to examine whether the training converges. Both models were trained on NVIDIA A100 GPUs, and the training converged in 60 epochs.

## 4 EXPERIMENTS

In this section, we evaluate VER on definition modeling, relation modeling, and hyper-relation modeling in three settings: 1) fine-tune the model on the full task-specific training data; 2) fine-tune the model in low-resource settings; 3) directly use the model without fine-tuning. The main goal of the experiments is to verify whether the continual pre-training step can enhance models' ability on verbalizing entities and relations without relying on external knowledge.

### 4.1 EXPERIMENTAL SETUP

**Datasets**. For definition modeling, we use the datasets presented in (Huang et al., 2021b) (**CS**, **Math**, **Phy**) and (August et al., 2022) (**DC**). For relation modeling, we use the dataset built in (Huang et al., 2022a) (**ORM**), we take the filtered test set for evaluation since the quality is higher. For hyper-relation modeling, there is no existing dataset. We find that **CommonGen** (Generative Commonsense Reasoning) (Lin et al., 2020) can serve our purpose for evaluation since the task formulation is similar: given a set of common concepts, i.e., $\mathcal{E}$ ($3 \le |\mathcal{E}| \le 5$), generating a coherent sentence describing an everyday scenario using these concepts. By testing on CommonGen, we can also measure the ability of our model for domain adaptation. Since the reference sentences of the official test set of CommonGen are not released, for the full data setting, we submit the results generated by the model to the leaderboard to get the performance. For the low-resource settings, we use the in-house split presented in (Wang et al., 2022) to facilitate comparison between our model and the baseline.

**Baselines**. Since VER is trained based on BART (Lewis et al., 2020), we compare VER with BART. It is not our purpose to advocate our model can achieve state-of-the-art performance on the specific tasks (they are usually achieved by leveraging external knowledge such as a knowledge graph. More details are in Section 5). What we want to verify is whether the continual pre-training step can enhance the ability of the model on verbalizing entities and relations. We may change the base model to other models such as T5 (Raffel et al., 2020) and continually train the model in a similar manner to achieve a different performance. Furthermore, other methods may replace their base model (e.g., BART, if used) with VER to achieve better performance (e.g., VER + A* Neurologic (Lu et al., 2022)).

**Metrics**. For definition modeling and relation modeling, we follow Huang et al. (2021b; 2022a) to use BLEU (BL) (Papineni et al., 2002)[6], ROUGE-L (R-L) (Lin, 2004), METEOR (MT) (Banerjee & Lavie, 2005), and BERTScore (BS) (Zhang et al., 2019) for automatic evaluation. Among them, BLEU, ROUGE-L, and METEOR focus on measuring surface similarities by n-gram overlap, and BERTScore is based on the similarities of contextual token embeddings. For the evaluation on generative commonsense reasoning, we follow Lin et al. (2020) to use BLEU-4, CIDEr (Vedantam et al., 2015), and SPICE (Anderson et al., 2016), where CIDEr and SPICE focus on evaluating the concept association instead of n-gram overlap.

**Implementation details**. For each task, to make the results comparable and reproducible, we adopt the same hyperparameters as the implementation of the authors to fine-tune BART. We also use the same hyperparameters as BART to fine-tune VER on specific tasks. For definition modeling, since Huang et al. (2021b) use BART-base, we fine-tune VER-base for a fair comparison. For relation modeling and generative commonsense reasoning, we use VER-large. For the low-resource settings,

---

[5] `https://github.com/facebookresearch/fairseq`

[6] The version implemented on `https://github.com/mjpost/sacrebleu`.

Table 1: Results of definition modeling.

| *100%* | CS | | | | Math | | | | Phy | | | | DC | | | |
|---|---|---|---|---|---|---|---|---|---|---|---|---|---|---|---|---|
| | BL | R-L | MT | BS | BL | R-L | MT | BS | BL | R-L | MT | BS | BL | R-L | MT | BS |
| BART | 8.31 | 28.02 | 12.83 | 77.97 | 6.89 | 28.50 | 10.97 | 76.45 | 5.28 | 25.75 | 10.57 | 76.88 | 13.13 | 31.75 | 13.30 | 79.31 |
| VER | 8.43 | 30.11 | 13.06 | 79.57 | 7.09 | 31.94 | 11.86 | 78.07 | 7.09 | 30.63 | 12.71 | 79.18 | 13.95 | 33.57 | 14.84 | 80.49 |
| *10%* | BL | R-L | MT | BS | BL | R-L | MT | BS | BL | R-L | MT | BS | BL | R-L | MT | BS |
| BART | 3.50 | 22.98 | 8.68 | 75.55 | 4.32 | 25.42 | 8.94 | 75.21 | 3.27 | 24.19 | 8.43 | 75.72 | 5.56 | 23.97 | 9.47 | 77.13 |
| VER | 6.43 | 28.24 | 12.36 | 78.77 | 7.24 | 31.18 | 11.79 | 77.82 | 6.43 | 30.57 | 12.42 | 78.92 | 7.59 | 28.25 | 12.09 | 78.70 |
| *0%* | BL | R-L | MT | BS | BL | R-L | MT | BS | BL | R-L | MT | BS | BL | R-L | MT | BS |
| VER | 5.05 | 26.55 | 11.96 | 77.84 | 6.33 | 30.36 | 11.57 | 76.88 | 5.95 | 28.79 | 12.35 | 78.13 | 6.06 | 23.49 | 11.22 | 75.49 |
| VER$^-$ | 4.81 | 26.24 | 11.62 | 77.55 | 6.00 | 30.57 | 11.41 | 77.35 | 5.70 | 28.62 | 12.12 | 78.06 | 5.98 | 22.84 | 11.01 | 75.32 |

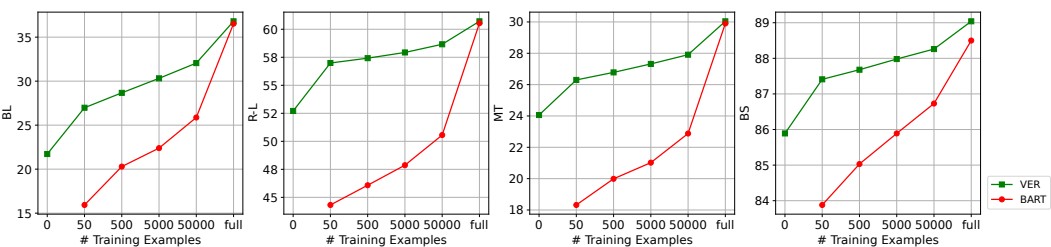

Figure 3: Results of open relation modeling (ORM) with different numbers of training examples.

we randomly sample the corresponding number of training samples from the train sets. For all the models and settings, we train the models with enough epochs to ensure the training converges and select the checkpoint with the best validation performance.

## 4.2 DEFINITION MODELING

In Table 1, we report the results of definition modeling on the four datasets. For the full-data setting (*100%*), VER outperforms BART on all the datasets. In the low-resource setting (*10%*, i.e., fine-tune the model with only 10% of the data), we observe that VER achieves a more significant improvement. The results demonstrate that after continually pre-training the model with the entity(s)-to-sentence reconstruction task, the model acquires more knowledge about entities and has a better ability to verbalize entities.

Since VER can generate a sentence (possibly a definitional sentence) by taking any entity as input without fine-tuning, we also report the "0%" results, where no training data on the specific tasks are used to fine-tune the model. We find that VER (*0%*) can achieve better performance than BART (*10%*), which indicates the strong performance of VER on definition modeling without additional training.

To validate whether the joint training of relation modeling will benefit or harm the performance of definition modeling, we train a version of VER only with data examples where $|\mathcal{E}| = 1$ (VER$^-$). From the results in Table 1, we observe that the performance of VER$^-$ is slightly worse than VER, which means relation understanding by relation modeling and hyper-relation modeling can benefit (at least does not harm) definition modeling.

## 4.3 (OPEN) RELATION MODELING

Figure 3 summarizes the results of open relation modeling. We observe that VER consistently outperforms BART on the four metrics, and the performance improvement is more significant when the model is fine-tuned on less training data. We also find that the performance of the model without any additional fine-tuning (*# Training Examples = 0*) is quite impressive. On two metrics (R-L and MT), the performance is even better than BART trained with 50,000 examples.

Table 2: Sentences produced by BART and VER fine-tuned on ORM. VER (500) refers to VER fine-tuned with 500 training examples.

| Entities | | {*evaluation*, *computer science*} |
|---|---|---|
| BART | (0) | uationil.uation; |
| VER | (0) | The term "***evaluation***" is used in ***computer science*** to refer to the process of ***evaluation*** of a system. |
| BART | (500) | ***Computer science*** is the study of ***computer science.*** |
| VER | (500) | ***evaluation*** is a term used in ***computer science*** to describe the process of evaluating a program or system. |
| BART | (full) | In ***computer science, evaluation*** is the process of evaluating a computer program to determine whether it is suitable for the task at hand. |
| VER | (full) | In mathematics and ***computer science, evaluation*** is the process of determining whether a statement is true or false. |

| Entities | | {*machine learning*, *algorithm*} |
|---|---|---|
| BART | (0) | Machine***Machine Learning*** |
| VER | (0) | The learning ***algorithm*** is an ***algorithm*** used in the development of artificial neural networks. |
| BART | (500) | Deep learning is a type of ***machine learning algorithm***. |
| VER | (500) | In ***machine learning***, a learning ***algorithm*** is a class of ***machine learning algorithm***s that can be used to learn from data. |
| BART | (full) | ***Machine learning*** (ML) is a subfield of ***machine learning*** concerned with the design of ***algorithm*** s that can learn and learn better over time. |
| VER | (full) | ***Machine learning*** (ML) is a subfield of artificial intelligence that studies ***algorithm*** s that learn how to make predictions based on observed data. |

The results indicate that rich entity and relational knowledge are learned by VER through continual pre-training. Besides, the text reconstruction task enables the model to produce natural language representations of relations by connecting entities in a coherent sentence. The robustness of VER in low-resource settings also justifies the benefits of conducting the continual pre-training step.

Table 2 provides some generation examples of the models. For {*evaluation*, *computer science*}, VER without any fine-tuning can generate a reasonable sentence describing what *evaluation* means in *computer science*, while BART pre-trained on 500 training examples cannot produce a meaningful sentence. For {*machine learning*, *algorithm*}, the sentence produced by VER (0) is not very meaningful; however, after fine-tuning on the task-specific data (even with only 500 examples), the model can generate a coherent sentence describing the relationship between *machine learning* and *algorithm*, while the sentence generated by BART trained with full data still does not make sense.

## 4.4 HYPER-RELATION MODELING (GENERATIVE COMMONSENSE REASONING)

Table 3 reports the CommonGen leaderboard results of VER compared to BART. We find that although the style of sentences used to pre-train VER is quite different from that in CommonGen, e.g., "A *dog* leaps to *catch* a *thrown frisbee*", the continual pre-training step still benefits the generative commonsense reasoning ability of the model.

From the results of low-resource experiments in Figure 4, we observe that the improvement of VER in the low-resource settings is very significant, despite the style of sentences in the pre-training and fine-tuning being quite different. Although the zero-shot performance of VER on CommonGen is poor, with 50 training examples, VER can achieve better results than BART trained with 5,000 training examples on CIDEr and SPICE (according to (Lin et al., 2020), SPICE and CIDEr correlate the human evaluation the most). This indicates the domain adaptation ability of VER is high.

In Table 4, we present some sample outputs of the models. Here {*dog*, *frisbee*, *catch*, *throw*} is an example in the test set of CommonGen that is used as the demonstration in their paper. From the results, we find that despite all the baselines failing in this example, VER produces a plausible sentence describing a correct everyday scenario using all the concepts. We also find that the performance of VER in low-resource settings is very impressive. For instance, BART trained with 5,000 training examples cannot even generate a sentence containing all the concepts, and the generated

Table 3: Results on CommonGen (leaderboard v1.1).

|      | BLEU-4 | CIDEr | SPICE |
|------|--------|-------|-------|
| BART | 31.83  | 13.98 | 28.00 |
| VER  | 34.22  | 16.28 | 28.28 |

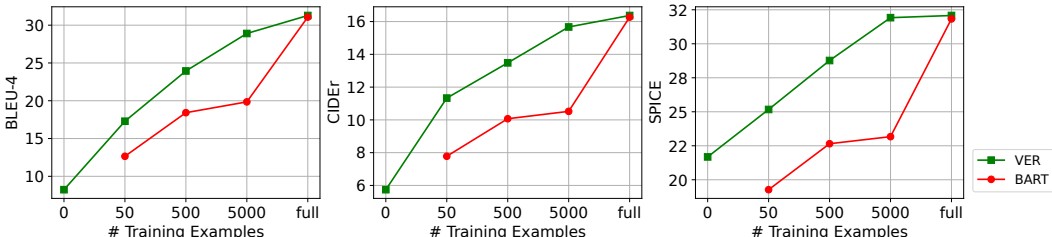

Figure 4: Results of the low-resource experiments on CommonGen (in-house) with different numbers of training examples.

Table 4: Sentences produced by commonly-used pre-trained models and VER fine-tuned on CommonGen. VER (50) refers to VER fine-tuned with 50 training examples. Here we take the example in the demonstration of (Lin et al., 2020).

| Concepts | | {*dog*, *frisbee*, *catch*, *throw*} |
|----------|---|-------------------------------------|
| Human 1 | | A **dog** leaps to **catch** a **thrown frisbee**. |
| Human 2 | | The **dog catches** the **frisbee** when the boy **throws** it. |
| Human 3 | | A man **throws** away his **dog**'s favorite **frisbee** expecting him to **catch** it in the air. |
| GPT-2 | | A **dog throws** a **frisbee** at a football player. |
| UniLM | | Two **dogs** are **throwing frisbees** at each other . |
| BART | | A **dog throws** a **frisbee** and a **dog catches** it. |
| T5 | | **dog catches** a **frisbee** and **throws** it to a **dog** |
| VER | | A man is **throwing** a **frisbee** to his **dog**, who **catches** it. |
| BART | (0) | ;; |
| VER | (0) | a **dog** that is trained to **throw** and retrieve a **frisbee** by its handler is given the task of making a **catch** and **throw** of the disc. |
| BART | (50) | A boy is playing **frisbee** with his friends |
| VER | (50) | a **dog catches** a **frisbee** and **throws** it to a person. |
| BART | (500) | A **dog catches** a **frisbee** during a football game. |
| VER | (500) | A **dog catches** a **frisbee** and **throws** it. |
| BART | (5000) | A man is **throwing** a **frisbee** to a woman who is **catching** it. |
| VER | (5000) | Two **dogs** are playing **frisbee** and one of them is **catching** and **throwing** it. |

sentence describes a weird scenario, while VER trained with only 50 examples can generate a coherent sentence containing all the concepts. Besides, without fine-tuning, BART cannot generate anything meaningful, while VER can still generate a reasonable sentence using all the concepts, although the style of the sentence is different from the ground truth.

## 5 RELATED WORK

**Definition Modeling**. *Definition modeling* aims to generate a definition for a given entity/term. This problem was first studied in Noraset et al. (2017) in a form of generating definitions of words with word embeddings. Later works include Gadetsky et al. (2018); Ishiwatari et al. (2019); Washio et al. (2019); Mickus et al. (2019); Li et al. (2020a); Reid et al. (2020); Bevilacqua et al. (2020); Huang et al. (2021a;b); August et al. (2022). For instance, Bevilacqua et al. (2020) fine-tune BART Lewis et al. (2020) on the word/phrase-definition pairs with given contexts. August et al. (2022)

aims to control the complexity of the definition while generating the definition for a given term. Huang et al. (2021b) propose to combine definition extraction and definition generation to improve the performance of definition modeling.

**(Open) Relation Modeling**. *Open Relation Modeling* (Huang et al., 2022a) aims to generate a sentence to describe the relationship within a given entity pair. The authors propose to fine-tune BART and incorporate reasoning paths in knowledge graphs as auxiliary knowledge to solve this task. As follow-up work, Huang et al. (2022b) construct *Descriptive Knowledge Graph* by extracting and generating sentences explaining entity relationships with the analysis of dependency patterns and a transformer-based relation description synthesizing model.

**Generative Commonsense Reasoning.** *Generative Commonsense Reasoning* (Lin et al., 2020) is a constrained text generation task that tests machines' ability to generate a coherent sentence describing everyday scenarios containing the given concepts. Later works mainly focus on improving performance by retrieving external knowledge to help the generation. For instance, KG-BART (Liu et al., 2021) designs a knowledge graph-augmented model that incorporates the embeddings of relations of concepts from ConceptNet (Speer et al., 2017) as auxiliary inputs of BART. EKI-BART (Fan et al., 2020), Re-T5 (Wang et al., 2021), and KFCNet (Li et al., 2021) retrieve prototype sentences from external corpora as auxiliary input to language models such as BART and T5 (Raffel et al., 2020). Although these methods enable the model to achieve better performance by utilizing those auxiliary information, they do not really improve the model's ability of generative commonsense reasoning (*Note*: the original purpose of CommonGen benchmark is "to explicitly test machines for the ability of generative commonsense reasoning" (Lin et al., 2020)). In this work, we show that continual training on verbalizing entities and relations, rather than incorporating auxiliary information in the input, can truly improve models' generative commonsense reasoning ability.

**Natural Language Representation**. In this paper, we define natural language representation as a text description of entities and relations. The concept can be further extended to other tasks. For instance, image captioning (Stefanini et al., 2021; You et al., 2016; Cornia et al., 2020; Li et al., 2020b) can be considered as learning natural language representations of images, document summarization (Allahyari et al., 2017) can be regarded as learning shorter natural language representations of documents. Since the surface name of entities does not provide much semantic knowledge, different from previous works where the representation can be translated or extracted from the input, natural language representations of entities and relations challenge the model's ability to generate coherent sentences describing entities and relations based on its reasoning ability and knowledge stored in the parameters of the model.

## 6 CONCLUSION

In this paper, we propose **VER**: A Unified Model for **V**erbalizing **E**ntities and **R**elations. We combine definition modeling, relation modeling, and hyper-relation modeling in a unified form and pre-train VER on a large training data by forming the "entity(s) → sentence" reconstruction task. Extensive experiments on three tasks and six datasets demonstrate the superiority of our model, especially in low-resource settings.

There are various applications of VER. First, VER itself can be used as a tool for humans to explore entities and relations by providing interpretable text descriptions, which can help humans better understand entities and relations. This is particularly useful in the scientific domain, where researchers come across new terms every day and want to understand previously unknown concepts and relationships between relevant concepts, and in the e-commerce domain, where users want to understand the function of specific products and the relationship between the recommended product and the product he/she already bought (e.g., *tripod* and *camera*). Second, as shown in our experiments, VER can be applied to improve the performance on entity and relation verbalization tasks such as definition modeling, relation modeling, and generative commonsense reasoning. Third, VER can serve as a knowledge source to provide knowledge on entities and relations to enhance models designed for entity&relation-related tasks such as entity typing Ren et al. (2016), relation extraction (Bach & Badaskar, 2007), and knowledge graph completion (Lin et al., 2015), which we leave as future work for us and the whole research community.

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
