# OpenReview forum: "VER: Learning Natural Language Representations for Verbalizing Entities and Relations"
_ICLR.cc/2023/Conference — Submitted to ICLR 2023_

### Official Review · Reviewer_Fgdc · 2022-10-19

**Confidence:** 4
**Correctness:** 3
**Technical Novelty And Significance:** 2
**Empirical Novelty And Significance:** 2
**Recommendation:** 3

**Clarity, Quality, Novelty And Reproducibility:**

Clarity is top-notch. The Reviewer has no complaints about it. A minor suggestion is to shorten Eq.1 to Eq.3 as they basically talk about the same thing.

Quality weakness is mostly in the evaluation, that a weak baseline was used, and the risk of hallucination was not covered.

Novelty is weak. The methodology is a small adaption of BART to a new seq2seq task. The setup is quite routine.

Reproducibility is good. The authors shared part of the model and promised to share the bigger one when published.

Minor suggestion: the authors might want to add a "pronounced as xxx" to the emoji-embedded name of the model.

**Strength And Weaknesses:**

Strength:

1) This paper is really well written. It's very easy to follow the author's thoughts, and all details (e.g training set up) have been explained clearly. The Reviewer didn't spot any obvious errors in writing.

Weakness:

1) The baseline is weak (BART). Given that VER is a model fine-tuned on BART for this task, it's almost guaranteed to be better performing. The Reviewer would like to see comparisons with STOA algorithms, even if VER is not surpassing them.
2) Unclear about generalization ability. The data for training (Wikipedia) is already a pretty comprehensive set of descriptive relationships of entities. It's not clear how the model could generalize, vs just remembering what appeared in Wikipedia text. It's not clear to the Reviewer about how much overlap do Wikipedia and the evaluation data have.
3) No discussion on hallucination. In order to make the results useful, e.g. helping human users or serve as a knowledge base, we need to understand how correct is VER's output. If for new (or even existing) entities, the model simply hallucinates to produce seemingly correct sentences, it would do more harm than help.

**Summary Of The Paper:**

This paper proposed a model to turn one or more entities into a natural language sentence that describes their relationship (or define the entity when there's just one). The model is a modified BART model trained with entity->sentence samples from Wikipedia. By doing a single seq2seq training, the model is able to perform definition and relation description tasks with one set of parameters.

Evaluation was done on open datasets of definition (DC), relation modeling (ORM) and hyper-relation modeling (CommonGen). The results were compared with vanilla BART model on an array of commonly used metrics. It showed that the proposed model (VER) outperforms BART in low-resource settings. When the training data is sufficient, their performance is on-par.

The paper argued that the model can be used as a tool for human users to explore new concept, as well as a base model for fine-tuning towards related tasks. The generated NL descriptions may serve as an interpretable  knowledge base.

**Summary Of The Review:**

The Reviewer thinks the paper is a well written one, very easy to follow. However, the result was not thoroughly tested (weak baseline) and discussed (hallucination risk). The methodology itself is straightforward and training setup is quite routine, with less discussion on design challenges.

The Reviewer would suggest adding stronger, STOA baselines and check the model's generalization abilities beyond Wikipedia, as well as assess the hallucination risk.

---

### Official Review · Reviewer_m8ZD · 2022-10-26

**Confidence:** 4
**Correctness:** 2
**Technical Novelty And Significance:** 2
**Empirical Novelty And Significance:** 2
**Recommendation:** 3

**Clarity, Quality, Novelty And Reproducibility:**

The paper is not very clear technically, and is not described in enough detail to be reproducible.
More detailed comments:
- If BART was pretrained on Wikipedia, it merely learns to extract sentences that are similar to the first sentence(s) on a Wikipedia page
(either by memorization, or based on style cues?). More discussion and error analysis might shed light on what the model is intended to learn, and what is learns in practice.
- "For each page, we extract the plain text by WikiExtractor3. And we use the neuralcoref (Clark & Manning, 2016) coreference resolution
tool in spaCy4 to preprocess the documents." - I can guess why, however the motivation to do so should be stated.
- Crucially, it is not clear what the difference is exactly between BART and VER (Table 1). Is BART also fine-tuned using
the same data, but with consecutive sentences?
- The paper should include some explanation about the datasets to make the paper self-contained.
- In Table 2, the full BART gives better sentence than VER. Can you refer to that?
- It would be interesting to compare the generated definition sentences with formal glosses.
- The results should compare with some other existing methods (which are mentioned in the related work section) on definition generation.

**Strength And Weaknesses:**

Pro:
- An interesting task
- A creative solution

Cons:
- The experimental setup is not clear
- The experiments are not comprehensive

**Summary Of The Paper:**

The work is about verbalizing entities and the relationships between them. According to the authors, existing
works deal with entity and relation verbalization separately. This work aims to solve entity and relation verbalization
in a unified form by adaptive generative language models to the task of entity and relation understanding.

Specifically, definition modeling and relation modeling are formulated uniformly as follows:
given a set of entities (or a single entity) in string form, a sentence is generated that describes the entities and their relationships.
When a single entity is given, it is equivalent to definition modeling. When the size of the set is 2, it is equivalent to relation modeling.

The proposed approach is called VER (a Unified Model for Verbalizing Entities and Relations):
- first, a large dataset is constrcuted from Wikipedia. The first (5) sentences from each page are included in the dataset, as those sentences
are often definitions of the entity or its relationships. Entity mentions are identified using Wikipedia hyperlinks.
The models are pre-trained using `self-supervised text reconstruction': given an entity or a set of entities, they are matched with the relevant sentences
in the training corpus.

The model of choice is BART (a transformer-based encoder-decoder architecture), using both BART-base and BART-large.
The following settings are evaluated: using full training data,
Evaluate three settings: fine-tuning on full training data, partial data (10%; a low-resource setup) and with no fine-tuning.

The experiments mainly consider datasets in scientific domains (CS, math, physics).
Also, the models are tested on CommonGen, where given multiple entities, the goal is to generate a coherent
sentence describing an everyday scenario using these concepts. The reference sentences of
the official test set of CommonGen are not released, and results are computed via a leaderboard.
The authors acknowledge however that state-of-the-art performance is achieved by leveraging external knowledge sources.

The results demonstrate that after continually pre-training the model with the
entity(s)-to-sentence reconstruction task, the model acquires a better ability to verbalize entities.
In the low-resource setup, VER achieves a more significant improvement compared with BART.

To validate whether the joint training of relation modeling will benefit or harm the performance of
definition modeling, a version of VER is trained only with data examples of a single entity. The results show that relation modeling
and hyper-relation modeling do not harm definition modeling.

**Summary Of The Review:**

The paper is not clear technically, and is not described in enough detail to be reproducible.

---

### Official Review · Reviewer_yzmq · 2022-10-28

**Confidence:** 4
**Clarity, Quality, Novelty And Reproducibility:** The work is clearly written, while ho…
**Correctness:** 3
**Technical Novelty And Significance:** 2
**Empirical Novelty And Significance:** 2
**Recommendation:** 3

**Strength And Weaknesses:**

Strengths:
- The results are impressively better than the baseline (BART)
- The dataset generation is interesting and quite interesting and relevant to the community – would this dataset be released to the public?

Weaknesses:
- I was not convinced with the motivation behind verbalizing entities and relations. The paper focuses the story in the beginning on how humans learn entities and relations, and then goes on to saying they have developed a method that can verbalize entities and relations the way humans do. However, humans do this to create a better model of the world for themselves. How is this verbalization useful to machines, if we continue with the analogy to humans? Furthermore, real applications to this method are not mentioned until the end of the introduction section. I would expect these to be mentioned much sooner in the paper.
- Results compare different generation metrics. However, we know how contentious these are in the literature right now. Could there be an explanation provided on how significant the performance improvements over the baseline BART are?
- I was expecting a comparison to Lin et al 2020 for the hyper relation modeling since that was mentioned at the end of the introduction section. However, the results once again only compare BART as a baseline for the hyper relation modeling (Table 3). You may correct me if I am looking at the wrong table for those results.
- For the dataset evaluation, can there be a discussion provided on the train vs test splits –  mainly to answer the question that if you are evaluating the model on generation for entities and relations, do you provide different sets of them for train vs test set ? Does the model run into memorization issues, similar to how many generation models do, and if so, how much? Does the model generate different utterances semantically for the same entities and relations, across different runs?

**Summary Of The Paper:**

This paper proposes a new method for combining definition modeling, relation modeling and hyper-relation modeling towards the purpose of generating natural language utterances for verbalizing entities and relations. The contributions of the paper are in combining multiple techniques of entity and relation modeling and devising a new pre-training task for “entity(s)-->sentence” reconstruction.

**Summary Of The Review:**

I found the paper to be an interesting read, but was left with many unanswered questions as listed in the weaknesses section.

---

### Official Review · Reviewer_WEUx · 2022-10-30

**Confidence:** 2
**Correctness:** 3
**Technical Novelty And Significance:** 4
**Empirical Novelty And Significance:** 3
**Recommendation:** 6

**Clarity, Quality, Novelty And Reproducibility:**

The paper proposes a unified way of learning natural language representations of entities and relations. Extensive experiments on three tasks and six datasets demonstrate the superiority of the proposed method, especially in low-resource settings. The paper is clearly presented and easy to read. The idea of a unified way of learning natural language representations of entities and relations is novel.

**Strength And Weaknesses:**

Strength:
1) The paper proposes a fundamental and unified way to pre-train the models for learning natural language representations of entities and relations.
2) Extensive experiments on three tasks and six datasets demonstrate the superiority of the proposed method, especially in low-resource settings.

Weakness:
1) There are no comparisons between the proposed method and SOTA results in each task. It is understandable that the main contribution of the paper is about learning natural language representations of entities and relations in a unified way. However, adding comparisons between the proposed method and SOTA in an aligned way (e.g., without using external knowledge) can help the readers understand how the proposed method advances the SOTA performance in these tasks (or what is the performance gap between the proposed method and SOTA).
2) Since the contribution of the paper is to propose a unified way of learning natural language representations of entities and relations, it is interesting (and has more impact) if the paper can demonstrates improvements in more than one based models (e.g., T5).

**Summary Of The Paper:**

The paper connects definition modeling, relation modeling, and hyper-relation modeling in a unified form. The paper pre-trains VER (A
Unified Model for Verbalizing Entities and Relations) on a large training data by forming the “entity(s) → sentence” reconstruction task, which makes VER a useful tool for learning natural language representations of entities and relations.

**Summary Of The Review:**

The paper proposes a unified way of learning natural language representations of entities and relations. Extensive experiments on three tasks and six datasets demonstrate the superiority of the proposed method, especially in low-resource settings. There are no comparisons between the proposed method and SOTA results in each task. It is understandable that the main contribution of the paper is about learning natural language representations of entities and relations in a unified way. However, adding comparisons between the proposed method and SOTA in an aligned way (e.g., without using external knowledge) can help the readers understand how the proposed method advances the SOTA performance in these tasks (or what is the performance gap between the proposed method and SOTA). Besides, it is interesting (and has more impact) if the paper can demonstrates improvements in more than one based models (e.g., T5).

---

### Decision · Program_Chairs · 2023-01-20

**Decision:**

Reject

**Justification For Why Not Higher Score:**

The approach is mostly an adaptation of BART, and the baseline (BART) is rather weak.

**Justification For Why Not Lower Score:**

n/a

**Metareview: Summary, Strengths And Weaknesses:**

The paper  provides a framework (VER) for  pre-training models for learning natural language representations of entities and relations.
The proposed pretraining objective cleanly brings together different pieces and is novel.  The paper is very well-written.  However, there are concerns about motivation,  the incremental nature of the proposed approach in mostly adapting seq2seq LMs, and insufficient baselines.  The authors are encouraged to continue their work, address review concerns, and  resubmit at a future conference at ICLR or other venues.


**Summary Of Ac-Reviewer Meeting:**

n/a